# Nutritional Calcium Supply Dependent Calcium Balance, Bone Calcification and Calcium Isotope Ratios in Rats

**DOI:** 10.3390/ijms23147796

**Published:** 2022-07-14

**Authors:** Jeremy Rott, Eva Teresa Toepfer, Maria Bartosova, Ana Kolevica, Alexander Heuser, Michael Rabe, Geert Behets, Patrick C. D’Haese, Viktoria Eichwald, Manfred Jugold, Ivan Damgov, Sotirios G. Zarogiannis, Rukshana Shroff, Anton Eisenhauer, Claus Peter Schmitt

**Affiliations:** 1Center for Pediatric and Adolescent Medicine, Im Neuenheimer Feld 430, 69120 Heidelberg, Germany; jeremyrott@web.de (J.R.); teresa.toepfer@web.de (E.T.T.); maria.bartosova@med.uni-heidelberg.de (M.B.); michael.rabe@med.uni-heidelberg.de (M.R.); damgov@imbi.uni-heidelberg.de (I.D.); szarog@med.uth.gr (S.G.Z.); 2GEOMAR Helmholtz Centre for Ocean Research Kiel, 24148 Kiel, Germany; akolevica@geomar.de (A.K.); aheuser@geomar.de (A.H.); aeisenhauer@geomar.de (A.E.); 3Laboratory of Pathophysiology, Department of Biomedical Sciences, University of Antwerp, Campus Drie Eiken, Universiteitsplein 1, 2610 Wilrijk, Belgium; geert.behets@uantwerpen.be (G.B.); patrick.dhaese@uantwerpen.be (P.C.D.); 4Core Facility, Small Animal Imaging Center, German Cancer Research Center (DKFZ), 69120 Heidelberg, Germany; viktoria.eichwald@dkfz-heidelberg.de (V.E.); m.jugold@dkfz-heidelberg.de (M.J.); 5Renal Unit, University College London Great Ormond Street Hospital and Institute of Child Health, London WC1N 1EH, UK; rukshana.shroff@gosh.nhs.uk

**Keywords:** calcium, isotope, fractionation, calcium deficiency, bone mineralization

## Abstract

Serum calcium isotopes (δ^44/42^Ca) have been suggested as a non-invasive and sensitive Ca balance marker. Quantitative δ^44/42^Ca changes associated with Ca flux across body compartment barriers relative to the dietary Ca and the correlation of δ^44/42^Ca_Serum_ with bone histology are unknown. We analyzed Ca and δ^44/42^Ca by mass-spectrometry in rats after two weeks of standard-Ca-diet (0.5%) and after four subsequent weeks of standard- and of low-Ca-diet (0.25%). In animals on a low-Ca-diet net Ca gain was 61 ± 3% and femur Ca content 68 ± 41% of standard-Ca-diet, bone mineralized area per section area was 68 ± 15% compared to standard-Ca-diet. δ^44/42^Ca was similar in the diets, and decreased in feces and urine and increased in serum in animals on low-Ca-diet. δ^44/42^Ca_Bone_ was higher in animals on low-Ca-diet, lower in the diaphysis than the metaphysis and epiphysis, and unaffected by gender. Independent of diet, δ^44/42^Ca_Bone_ was similar in the femora and ribs. At the time of sacrifice, δ^44/42^Ca_Serum_ inversely correlated with intestinal Ca uptake and histological bone mineralization markers, but not with Ca content and bone mineral density by µCT. In conclusion, δ^44/42^Ca_Bone_ was bone site specific, but mechanical stress and gender independent. Low-Ca-diet induced marked changes in feces, serum and urine δ^44/42^Ca in growing rats. δ^44/42^Ca_Serum_ inversely correlated with markers of bone mineralization.

## 1. Introduction

The bone matrix, defined as bone without cells, consists of organic components, mainly collagen type 1, glycoproteins and proteoglycans and of inorganic structures, mainly hydroxyapatite [1]. The hydroxyapatite attaches in crystalline form to the collagen fibrils to render the structure hard and pressure-resistant. The main elemental components of hydroxyapatite are calcium (Ca) and phosphorous (P), which are supplied by the blood during bone formation, while bone resorption leads to Ca release into the circulation. The dynamic balance between bone formation and resorption is mainly controlled by Ca fluxes into the bone and out of the bone and can be described by a simple box model [2]. The bone mass is increasing when the ratio of bone Ca gain/Ca loss is above 1, decreasing when this ratio is below 1 and in equilibrium when it equals 1 [3], a status which is usually referred to as the “peak bone” interval [4]. Growing children require a positive Ca balance, most so during the first two years of life and during the pubertal growth spurt; peak bone mass is reached in average at the age of 25–30 years. In older individuals bone resorption usually outweighs bone formation [4].

Currently, bone formation and resorption can only be detected indirectly, i.e., by dual X-ray absorptiometry (DXA), by quantitative bone histomorphometry, which is rarely performed in clinical routine and by determination of serum bone markers reflecting osteoblast and osteoclast activity. However, up to now there is no biochemical parameter quantitatively reflecting the bone Ca mineralization process.

Of the naturally occurring Ca, 2.1% is ^44^Ca, 0.65% is ^42^Ca and 96.9% is ^40^Ca [5]. Recently, determination of serum and urine Ca isotope ratios (δ^44/42^Ca_Serum_ and δ^44/42^Ca_Urine_) has been proposed as a tool to quantify Ca fluxes between the different compartments of the body [6]. The stable Ca isotopes have a mass ranging from 40–48 atomic mass units, and independent of the mass, their chemical characteristics are the same. However, in incomplete kinetic reactions the relatively light isotopes are enriched in the product and the relatively heavier isotopes preferentially remain in the educt, since in chemical reaction light isotopes have a higher reaction rate than the heavy ones and overcome chemical barriers easier than heavier isotopes [5,7]. As a result, the isotopes separate from each other and thereby change the relative abundance, a process which is referred to as kinetic “mass-dependent isotope fractionation”. To express the magnitude of enrichment in light isotopes in the product, the measured isotopes are presented as a ratio of relatively heavy isotopes (^44^Ca) and relatively light isotopes (^42^Ca) compared to the ratio of a well-known international standard material such as NIST SRM 915a (NIST = National Institute of Standards and Technology) and is reported as δ^44/42^Ca per mil (‰) deviation relative to this standard [8].

Skulan and DePaolo demonstrated that mineralized tissue is enriched in light Ca isotopes compared to blood; light isotopes in serum are reduced with bone formation [2]. In vertebrates the Ca isotope difference between bone and serum is largely constant and about −0.3‰ [2,9]. Based on Ca tracer studies, the relaxation time of Ca in humans is about 200 days for the bones [10,11], but more than three orders of magnitude lower for serum, i.e., about one hour. Changes in the supply of Ca to the blood are reflected in the serum Ca isotope composition within about one hour, while bones respond to Ca changes much slower. Bed rest results in bone loss and a significant decrease in δ^44/42^Ca_Serum_ and δ^44/42^Ca_Urine_ occurs within 10 days [12]. In the case of osteoporosis, bone resorption chronically exceeds bone formation and leads to a decrease in δ^44/42^Ca_Serum_. The Ca isotope composition of the subsequently formed new bone also decreases to a value lower than before. Continued bone Ca loss exceeding bone Ca gain results in lighter δ^44/42^Ca ratio of both the bones and the serum; serum values are then indicative for bone Ca loss and eventually osteoporosis. This concept has been derived from several studies [3,6,12,13,14], suggesting that Ca isotope ratios may discriminate healthy from osteoporotic post-menopausal women. In the case of predominant bone Ca loss, the serum is enriched in the light isotopes shifting δ^44/42^Ca_Serum_ ratios below a well-defined threshold value indicating equilibrium between Ca input and output to the bones [6]. During predominant bone formation and bone calcification, such as in growing individuals, more light isotopes are extracted from the blood. δ^44/42^Ca_Serum_ ratios shifts towards values above the defined threshold. The urine, enriched in the remaining ^44^Ca, is excreted with increasing δ^44/42^Ca_Urine_ ratios [6,12,15]. While these observations are highly promising with regard to future application in various diseases states affecting bone calcification and bone health, the mutual quantitative δ^44/42^Ca fractionation steps between the different body compartments (intestine, blood circulation, bone and urine) and the impact of dietary Ca supply are uncertain. It is unknown whether δ^44/42^Ca differs between specific sites within one bone and between weight bearing and non-weight bearing bones and between genders. Similarly, the reflection of bone histology mineralization indices by δ^44/42^Ca_Serum_ has not yet been assessed in vivo. To address these crucial questions, we studied the Ca and δ^44/42^Ca kinetics in serum, urine and feces and the resulting bone Ca content and δ^44/42^Ca in young, growing rats on a standard and a Ca deficient diet.

## 2. Results

Animals from the three groups had similar body weights during the first two weeks (*p* = 0.23), and group 2 and 3 during weeks three to six (*p* = 0.40) (Appendix A). At sacrifice, body weight was 322 ± 12 g in group 1 and 470 ± 17 g and 479 ± 28 g in groups 2 and 3, respectively. Weekly body weight gain was particularly high during the first two weeks (Appendix A).

After two weeks, mean body length of animals in group 1 was 40.3 ± 0.6 cm. Body length gain during the four weeks of dietary intervention in group 2 and 3 relative to the body length obtained at the end of week two in group 1 was 5.3 ± 0.8 cm and 5.7 ± 1.0 cm (*p* = 0.24). Final body length was 45.4 ± 0.9 cm in group 2 and 45.9 ± 1.0 cm in group 3 (*p* = 0.15). Daily food intake, urine and fecal excretions were similar throughout the study between groups (Table 1). Serum Ca concentrations (Figure 1) were also similar throughout the study period: mean Ca concentrations in week one and two were 2.90 ± 0.03 mmol/L (group 1), 2.80 ± 0.21 mmol/L (group 2) and 2.84 ± 0.03 mmol/L (group 3; *p* = 0.39). During weeks three to six, mean serum Ca concentrations were 2.73 ± 0.10 mmol/L in group 2 and 2.65 ± 0.04 mmol/L in group 3 (*p* = 0.25). Serum creatinine, alkaline phosphatase and phosphate were normal throughout the study period, largely unaffected by diet (Appendix A).

### 2.1. Total Body Ca Balance

Ca concentrations were determined in the diet, urine and feces and the intestinal Ca uptake, and excretion via urine and feces were calculated (Table 1). The net daily Ca gain was calculated as *Ca uptake (food + water)—Ca excretion (feces + urine)*. Intestinal Ca absorption is given in absolute terms as *Ca uptake (food + water)—fecal Ca excretion* as well as the relative intestinal Ca absorption rate (absorbed Ca/oral Ca uptake). Net Ca gains were comparable between the three groups during the first two weeks. However, animals on a low-Ca-diet had a 39 ± 2% lower Ca gain in weeks three to four and 39 ± 4% lower in weeks five to six (both *p* < 0.0001). Compared to the animals on a standard-Ca-diet, animals on a low-Ca-diet in total had a 61 ± 3% lower Ca gain during weeks three to six. The lower dietary Ca content was partially compensated by a higher relative intestinal Ca absorption rate as compared to animals on standard-Ca-diet of 24 ± 4% (week three to four, *p* = 0.0009) and 22 ± 8% (week five to six, *p* = 0.001) and a reduced urinary Ca excretion of 16 ± 29% (week three to four, *p* = 0.32) and 62 ± 35% (week five to six, *p* = 0.08; Table 1).

### 2.2. Femur Ca Content

Compared to the mean baseline femur Ca content measured after two weeks in group 1, femur Ca content increased by 27.0 ± 16 mg in the low-Ca-diet group over four weeks and by 39.6 ± 15 mg in the standard-Ca-diet group, i.e., the increase was only 68 ± 41% of the increase in femur Ca content in animals on standard-Ca-diet (Table 2). After four weeks of a low-Ca-diet, the Ca content per femur and the relative Ca content per bone mass were 10 ± 13% and 9.8 ± 14% lower than in rats maintained on standard-diet (*p* = 0.02/*p* = 0.03).

### 2.3. Tibia Micro Computed Tomography (µCT)

Tibia bone volume after four weeks on low-Ca-diet was compared to the tibia volume achieved with standard-Ca-diet (*p* = 0.0002). Mean bone volume gain compared to baseline (group 1, week 2) was 31 ± 23% lower with a low-Ca-diet (*p* = 0.0002). This was due to a smaller increase in cortical bone volume (*p* < 0.0001), while the cancellous bone volume was similar (*p* = 0.41; Appendix A). The volume differences were more pronounced in the metaphysis than in the diaphysis and epiphysis.

Bone density measurements by µCT closely correlate with values obtained from DXA [16,17] and were therefore used to quantify bone mineral density (BMD). Total bone density increased during the four weeks of dietary Ca intervention compared to baseline (group 1, week two), but in animals on low-Ca-diet the increase was only 80 ± 29% of the increase in animals on standard-Ca-diet (*p* = 0.03). The differences in bone density were distinct in the diaphysis, metaphysis and epiphysis, with the highest bone density values in the diaphysis. Taking the changes in tibia volumes into account, the mean increase in bone density × tibia volume (∆ Hounsfield Units × mm^3^ tibia volume) in animals on low-Ca-diet was 73 ± 25% of the increase in animals on a standard-Ca-diet (*p* = 0.001; Appendix A).

### 2.4. Tibia Histomorphometry

Quantitative histomorphometry of the left tibia demonstrated major bone alterations with low-Ca-diet. Total bone area and trabecular bone number were reduced, mineralized area per total tissue section area was reduced and respective osteoid area increased in animals on low-Ca-diet (Table 3). Osteoblast perimeter per osteoid perimeter as well as per total bone perimeter were increased, while osteoclast perimeter per either eroded perimeter or total bone perimeter remained unchanged compared to rats receiving the standard-Ca-diet. The mineralized area per section area in rats on a low-Ca-diet was only 68 ± 15% of the mineralized area in rats on a standard-Ca-diet (*p* = 0.001), mirroring the differences in net total body Ca uptake during the four weeks of intervention (Table 1), the differences in femur Ca content (Table 2), and µCT findings (Appendix A).

### 2.5. δ^44/42^Ca Isotope Studies

#### 2.5.1. δ^44/42^Ca in Serum, Feces and Urine

δ^44/42^Ca_Diet_ was 0.58 ± 0.06‰ in the low and 0.57 ± 0.08‰ in the standard-Ca-diet. Serum Ca concentration were similar between groups over time (Figure 1). During the two weeks of baseline, δ^44/42^Ca_Serum_ was similar in all three groups (0.52 ± 0.03, 0.53 ± 0.07, and 0.54 ± 0.06‰ in group 1 to 3; *p* = 0.91).

In animals on standard-Ca-diet, δ^44/42^Ca_Serum_ did not increase from the end of the baseline period (week two) until week 6 (*p* = 0.22), while in animals on low-Ca-diet δ^44/42^Ca_Serum_ increased (*p* = 0.007) and reached significantly higher values at week five (0.64 ± 0.07‰) and week six (0.63 ± 0.07‰) as compared to the δ^44/42^Ca_Serum_ at week two (*p* = 0.01/0.04). δ^44/42^Ca_Serum_ at sacrifice did not correlate with the bone Ca content.

Intestinal Ca absorption was significantly higher in animals on a low-Ca-diet (Table 1). The fecal Ca concentrations were similar at baseline (week 1–2) in the three groups (*p* = 0.80) and declined in the low-Ca-diet group during week three to week six (*p* = 0.01 vs standard-Ca-diet group) and was accompanied by higher absolute and relative intestinal Ca absorption (both *p* < 0.0001 versus animals on standard-Ca-diet; Table 1, Figure 2A). δ^44/42^Ca_Feces_ was similar in the three groups at baseline (*p* = 0.19), slightly declined in the animals on standard-Ca-diet during the subsequent four weeks (*p* = 0.001), and markedly declined in animals on low-Ca-diet (*p* < 0.0001). δ^44/42^Ca_Feces_ was much lower during week three to six in animals on low versus standard-Ca-diet (*p* < 0.0001; Figure 2B). δ^44/42^Ca_Feces_ increased with increasing relative intestinal Ca absorption in animals on standard-Ca-diet from weeks one to six (r = 0.49, *p* = 0.029; n = 20). Below an absorption rate of about 95%, almost all δ^44/42^Ca_Feces_ values were lower than their diet in animals on low-Ca-diet and inversely correlated from weeks three to six with relative intestinal Ca absorption (r = −0.68, *p* = 0.0015; n = 19; Figure 3).

Urinary Ca concentrations were lower in animals on the low-Ca-diet as compared to those receiving the standard-Ca-diet (*p* = 0.03; Figure 2C).

δ^44/42^Ca_Urine_ slightly declined in rats on the standard-Ca-diet from week 1 to 6 (*p* = 0.02) and even more in animals when switched to a low-Ca-diet for 4 weeks (*p* < 0.0001), along with a reduced urinary Ca excretion rate (Table 1). Moreover, δ^44/42^Ca_Urine_ was significantly lower in rats on low-Ca-diet than in respective animals on standard-Ca-diet (*p* < 0.0001). In animals on standard-Ca-diet, mean δ^44/42^Ca_Urine_ inversely correlated with mean δ^44/42^Ca_Serum_, but not in animals on low-Ca-diet (Appendix A).

#### 2.5.2. δ^44/42^Ca in Bone

Weight-bearing femora and non-weight-bearing ribs were homogenized and δ^44/42^Ca_Bone_ was determined. δ^44/42^Ca_Bone_ was similar in both types of bone, i.e., the Ca composition was not modified by bone function (Table 4-(1)), but increased after four weeks on standard-Ca-diet (*p* = 0.0008 and *p* = 0.0003 for femora and ribs, respectively), i.e., relative more ^44^Ca accumulated in the growing bone.

At week six, the increase in femur Ca content was smaller in rats on low-Ca-diet than in rats on standard-diet (Table 2), but relatively more heavy Ca isotopes accumulated in the femur and rib of animals on low-Ca-diet (Table 4-(1)). Altogether, animals on low-Ca-diet developed a relative increase of ^44^Ca in the serum (Figure 1), together with a relatively higher intestinal absorption and tubular reabsorption of ^44^Ca (Figure 2) and accumulated relatively more of the ^44^Ca in the femora and ribs than rats on the standard-Ca-diet.

To assess the Ca isotope distribution at different sites of a bone, δ^44/42^Ca_Bone_ was determined at predefined sites in the right tibia diaphysis, metaphysis and epiphysis. The δ^44/42^Ca_Bone_ was lower in the diaphysis than in the in metaphysis and epiphysis (Table 4-(2)). Female rat δ^44/42^Ca_Bone_ was similar at all bone sites studied (Appendix A).

Mean weekly δ^44/42^Ca_Serum_ varied in animals on low-Ca-diet, while in animals on standard-Ca-diet it was similar between animals, despite a larger range of intestinal Ca absorption (Figure 3). Overall mean weekly δ^44/42^Ca_Serum_ during week three to six inversely correlated with the net intestinal Ca uptake during this period (r = −0.65, *p* = 0.007; n = 16, Figure 4), a finding which could be reconfirmed in the subanalysis of the two distinct groups only in animals on low-Ca-diet (r = −0.61, *p* = 0.08; n = 9), but not in animals on standard-Ca-diet (r = 0.12, *p* = 0.79; n= 7). δ^44/42^Ca_Serum_ at time of sacrifice correlated with osteoid area/bone area (r = 0.64, *p* = 0.0005; n = 25), and negatively with tibia bone area/tissue area (r = −0.53, *p* = 0.007; n = 25), and mineralized area/total area (r = −0.53, *p* = 0.006; n = 25), but not with tibia bone mineral density as measured by µCT (r = 0.17, *p* = 0.35; n = 33), absolute femur Ca content (r = −0.02, *p* = 0.92; n = 26) and the gain in femur Ca during weeks three to six (r = −0.11, *p* = 0.64; n = 20). In animals on a standard-Ca-diet (group 1 and 2) δ^44/42^Ca_Serum_ at time of sacrifice correlated with tibia osteoid area/bone area (r = 0.70, *p* = 0.002; n = 16), but not with bone area/tissue area (r = −0.24, *p* = 0.37; n = 16), mineralized area/total area (r = −0.25, *p* = 0.35; n = 16), tibia bone mineral density as measured by µCT (r = 0.20, *p* = 0.36; n = 22), absolute femur Ca content (r = −0.31, *p* = 0.24; n = 16) and the gain in femur Ca during weeks three to six (r = −0.40, *p* = 0.26; n = 10). At the standardized measurement points of the diaphysis, metaphysis and epiphysis, µCT BMD findings inversely correlated with δ^44/42^Ca_Bone_ obtained from the same bone site in all animals (r = −0.37; *p* = 0.0017; n = 71), and in animals on standard-Ca-diet (r = −0.38, *p* = 0.015; n = 41) and low-Ca-diet (r = −0.60, *p* = 0.0005; n = 41).

## 3. Discussion

δ^44/42^Ca_Serum_ values have been suggested as a novel biomarker of bone metabolism reflecting the actual bone calcification process, based on studies in various animal models [2,9] and in humans [3,6,18,19,20]. ^42^Ca is preferentially incorporated into the bone, whereas ^44^Ca becomes enriched in the blood and is excreted in the urine [3,6,13,14,18,19,20]. The δ^44/42^Ca_Serum_ and δ^44/42^Ca_Urine_ increase during bone formation and decreased with bone resorption and age, with the highest values in adolescent boys [21] and the lowest values in osteoporotic post-menopausal women [6]. δ^44/42^Ca_Serum_ positively correlates with serum biomarkers of bone formation and negatively correlates with bone resorption markers and cortical bone mineral density [21]. In vitro, we recently demonstrated preferential deposition of the light Ca isotope with extracellular matrix calcification by osteoblasts, similar as in other biomineralization processes, and preferential transport of ^42^Ca across a proximal tubular epithelial cell barrier, whereas the fractionation process across endothelial cells and enterocytes cells was low [22]. Taken together, the reported findings are highly promising with regard to the application of δ^44/42^Ca_Serum_ and δ^44/42^Ca_Urine_ as a biomarker of bone Ca balance. An understanding of the fractionation processes at the compartment transitions is relevant in Ca homeostasis and is needed for mathematical modelling of Ca balances in vivo. Moreover, it has not yet been shown in how far δ^44/42^Ca is a biomarker of histological indices of bone calcification and actual bone Ca content.

We studied young, rapidly growing rats, which undergo major, rapid changes in bone morphology and Ca content and allow for a sensitive analysis of associated δ^44/42^Ca kinetics in the different body compartments. Body weight and length gain were comparable in rats on standard- and low-Ca-diet and as expected from the literature [23]. We infer that the reduced dietary Ca content was partially compensated by an increased intestinal absorption and lower urinary Ca excretion, maintaining normal serum Ca concentrations throughout the study. Both diets contained native vitamin D3 to prevent a respective bias of vitamin D deficiency. Still, after four weeks of low-Ca-diet, total net body Ca gain was about one third lower than compared to animals on the standard-Ca-diet, which resulted in a one-third lower bone volume and bone Ca content. In line with this, bone histology demonstrated an equally lower bone area per tissue area and bone mineralized area per total bone area together with the major accumulation of osteoid. Bone µCT imaging yielded similar dimensions of dietary Ca dependent differences when bone density changes were related to the concomitant differences in bone volume. Thus, our animal model provided consistent Ca mass balance findings, proving the suitability of the model to study the Ca isotope fractionation process in vivo.

The δ^44/42^Ca_Diet_ value of the food used in our study was about 0.58‰, independent of the Ca content. This is higher than the δ^44/42^Ca of dairy and non-dairy products in the European diet, which were estimated to be approximately −0.58‰ and −0.31‰, respectively [24,25]. In a recent publication, average dietary δ^44/42^Ca was calculated in children based on a prospective three-day food diary to be −0.46‰ (−0.49‰ to −0.41‰) [21]. The high δ^44/42^Ca of the animal food is due to the inorganic origin of Ca products used in this study (personal communication Altromin, Lage, Germany). In contrast, the δ^44/42^Ca value is about 1‰ lower in biological non-dairy and lowest in dairy foods, based on the Ca isotope fractionation process in the food chain [25]. The δ^44/42^Ca_Serum_ values in the rats were about 0.50‰, which is higher than usually observed in humans, where values range from about –1‰ in postmenopausal women to up to 0.3‰ in children [21] due to the inorganic origin of the diet (see above). In a cross-sectional study in children, a positive correlation was found between δ^44/42^Ca_Serum_ and height Z-score and higher δ^44/42^Ca_Serum_ values at pubertal Tanner stage 4 than 5, i.e., during the pubertal growth spurt. These observations reflect the higher Ca accumulation rates during rapid longitudinal growth. Independent of the dietary Ca intake, our young rats grew 5–6 cm within four weeks, corresponding to a linearly extrapolated (theoretical) average annual growth of 60–72 cm per year, which is much faster than the average annual growth in humans during the pubertal growth spurt of about 5–10 cm. The high Ca deposition rate in newly formed bone, with preferential deposition of isotopically lighter ^42^Ca, together with the high δ^44/42^Ca_Diet_ readily explains the high δ^44/42^Ca_Serum_ in the rapidly growing rats.

The first compartmental border to consider in Ca homeostasis is the intestinal epithelium, which regulates the uptake of dietary Ca. The intestinal Ca uptake was associated with major changes in δ^44/42^Ca_Feces_. δ^44/42^Ca_Feces_ was about two times greater than δ^44/42^Ca_Diet_ and the δ^44/42^Ca_Serum_, suggesting the preferential intestinal uptake of ^42^Ca. In animals with reduced dietary Ca supply, however, more than 90% of the dietary Ca was absorbed, fecal Ca concentrations were low, and fecal δ^44/42^Ca decreased to values below δ^44/42^Ca_Diet_ and δ^44/42^Ca_Serum_. Lowering of the δ^44/42^Ca_Feces_ values cannot be explained by kinetic isotope fractionation because isotope values should become increasingly higher with increasing absorption rate—other mechanisms have to be considered. The absorption of inorganic nutrients (minerals, trace elements) mainly takes place in the upper small intestine by the active or diffusive process, respectively. The Ca uptake rate depends on the daily Ca supply and is highest when the Ca intake is low [26,27]. It is speculated that during low Ca supply conditions most of the Ca, if not all, is already absorbed in the small intestine and that the feces actually contain small if any dietary Ca at all and the remnant Ca is from endogenous sources like digestive fluids and mucus cells [28,29,30]. These inferences are supported by the earlier observations of Heuser et al., 2016 [9], demonstrating lower δ^44/42^Ca_Feces_ in Ca deficient Göttingen minipigs compared to animals with standard Ca supply. Heuser et al. estimated the Ca isotope composition of the digestive fluids to be considerably lower than the corresponding δ^44/42^Ca_Diet_ and δ^44/42^Ca_Serum_. These observations are in full support of our findings here, suggesting that the digestive fluids are isotopically low in order to reconcile the kinetic fractionation process with the measured δ^44/42^Ca_Feces_. Following this hypothesis, a similar inference must then be made for the mucus cells, however, no such data are presently available. Following this approach, the conceptual model must take into account that at high intestinal Ca uptake during low Ca supply conditions, the Ca originating from digestive fluids and mucus cells dominates at the end of the gastrointestinal tract and the formation of the Ca isotope composition of the feces.

Another factor potentially contributing to the marked changes in fecal δ^44/42^Ca are different routes of calcium transport across the intestinal cell barrier. Dietary Ca is absorbed via the paracellular and transcellular route. The active, ATP dependent transcellular Ca uptake preferentially occurs in the duodenum and upper jejunum [26,27], while passive paracellular Ca absorption occurs mainly in the distal jejunum and ileum [31,32,33,34]. The paracellular pathway depends on the permeability of the tight junctions and the electrochemical gradient between the two sides of the barrier, which is the driving force for passive, energy independent diffusion [27]. Thus, the light ^42^Ca isotope should more rapidly cross the epithelial barrier following the physiological laws of diffusion provided by the Stokes-Einstein equation and Fick’s first law of diffusion [22]. In animals on a low-Ca-diet, the subtotal Ca uptake may be achieved by increasing the energy dependent transcellular uptake, with less or even no ^42^Ca isotope selectivity [35]. In this case, more of ^44^Ca is absorbed, increases δ^44/42^Ca_Serum_ and reduces fecal δ^44/42^Ca, and contributes to the low fecal δ^44/42^Ca. Thus, intestinal Ca uptakes may be associated with the preferential uptake of ^42^Ca unless the dietary Ca supply is low and compensatory energy dependent, non-isotope selective Ca uptake mechanisms are active.

δ^44/42^Ca_Urine_ values were two- to three-fold higher in animals on a standard-Ca-diet than δ^44/42^Ca_Serum_ values and were inversely correlated. This suggests the preferential tubular reabsorption of ^42^Ca and the enrichment of ^44^Ca in the urine, similar to recently reported findings in humans [21]. Rats on low-Ca-diet, however, had lower urinary Ca excretion together with lower δ^44/42^Ca_Urine_ ratios and the latter were not correlated with δ^44/42^Ca_Serum_. Renal tubular Ca reabsorption and δ^44/42^Ca handling follows similar principles as intestinal Ca uptake. Ca is freely filtered across the glomerular barrier, only a small fraction (∼1–2%) of filtered Ca is excreted under normal physiological conditions in healthy kidneys [36]. Seventy percent of the filtered Ca is passively reabsorbed across the paracellular route in the proximal tubule and the thick ascending limb, while active reabsorption across epithelial cells is needed in the distal convoluted, the connecting tubules and the collecting duct [36,37]. The concomitant decline in urinary δ^44/42^Ca with reduced urinary Ca excretion may be related with relatively higher tubular uptake of heavy Ca isotopes via the energy dependent Ca reabsorption via the transcellular route, with reduced Ca isotope selectivity.

Bone mineralization, i.e., extracellular matrix calcification, is associated with δ^44/42^Ca isotope fractionation, i.e., with preferential kinetic deposition of the lighter ^42^Ca isotope, similar to other biomineralization processes [22]. The δ^44/42^Ca gradient between bone and serum (δ^44/42^Ca_Bone_—δ^44/42^Ca_Serum_) in the rats was −0.44 ± 0.07‰ at week 2 and −0.36 ± 0.06‰ and −0.36 ± 0.08‰ with a standard- and low-Ca-diet, respectively. This is in the range of findings recently demonstrated in vitro [22] and in various mammalians, including humans [2,9,19]. In addition, we now demonstrate that δ^44/42^Ca is similar in weight-bearing and non-weight-bearing bones, independent of the age and the gender of the animals and the dietary Ca supply. Thus, conclusions drawn based on δ^44/42^Ca from serum and urine should apply independent of the mechanical stress, as different bones are exposed. Within the bones, δ^44/42^Ca distribution, however, differs, with lower ratios in the diaphysis, independent of the gender and the dietary Ca supply. The bone Ca isotope ratios inversely correlate with bone mineral density assessed by µCT at the different bone sites, with the highest HU counts in the diaphysis in all groups, the site of the largest mineralization, followed by fairly similar values in the metaphysis and epiphysis. Although the growth plate between the epiphysis and metaphysis is the site of Ca deposition during growth, the site of the highest amount of mineralization is the cortical bone of the diaphysis. Our findings support the preferential deposition of light Ca isotopes with bone mineralization as reported previously [2,3,6,38].

We then analyzed the relation of δ^44/42^Ca_Serum_ with bone histomorphological parameters of bone mineralization, using methods applied in the field of renal osteodystrophy, which also include the quantification of non-mineralized osteoid. The unmineralized osteoid is deposited by the osteoblasts, and serves as a ‘scaffold’ for the later mineralization. A recent study demonstrated a positive correlation of bone mineral density with δ^44/42^Ca of the metaphyseal tibia in rats with type 1 diabetes, but not in rats with chronic kidney disease and healthy rats [39]. In a larger number of healthy rats we now provide detailed information. In rats on low and standard-Ca-diet, δ^44/42^Ca_Serum_ inversely correlated with the mineralized bone area per total bone area and the bone area per tissue area and positively with the amount of non-mineralized osteoid area in the bone, i.e., the higher the δ^44/42^Ca_Serum_, the lower the bone mineralization activity. δ^44/42^Ca_Serum_ negatively correlated with the intestinal uptake of Ca, i.e., the lower the intestinal Ca uptake the higher the δ^44/42^Ca_Serum_. A low-Ca-diet resulted in subtotal intestinal Ca uptake and a δ^44/42^Ca_Feces_ even below δ^44/42^Ca_Serum_, a finding distinct from the δ^44/42^Ca_Feces_ in rats on a standard-Ca-diet, even though a subgroup of these rats had similarly high relative intestinal Ca uptake as rats on a low-Ca-diet. In the same direction, urinary Ca losses and δ^44/42^Ca_Urine_ were reduced in the animals on a low-Ca-diet, and δ^44/42^Ca_Serum_ increased. As illustrated in Figure 4, the negative correlation of δ^44/42^Ca_Serum_ with bone mineralization in the entire group of rats was based on the rats on the low-Ca-diet. No correlations of δ^44/42^Ca_Serum_ with histological markers of bone mineralization could be demonstrated in the animals on a standard-Ca-diet. Thus, the major changes in Ca isotope balances in the subgroup of animals on a low-Ca-diet may have overcome the effect of preferential deposition of the light ^42^Ca during bone mineralization. At all bone sites studied, diaphysis, metaphysis and the epiphysis of the tibia, and in the ribs and femur (Table 4), δ^44/42^Ca_Bone_ was higher the in the animals on low-Ca-diet than in rats on a standard-Ca-diet. These animals had quantitatively less bone Ca deposition, with relatively less preferential δ^42^ Ca isotope deposition from a serum pool with a higher δ^44/42^Ca.

Our assumption, that dietary Ca dependent differences in intestinal and renal δ^44/42^Ca isotope handling overrides the effects in δ^44/42^Ca_Serum_ by bone calcification, cannot be proven by our studies. Another limitation regards the histomorphometric analyzes. They were performed in the trabecular bone compartment of the tibia, the metabolically most active compartment, and not in the cortical bone, which is less metabolically active but contains a relatively higher amount of Ca, while Ca quantification involved both the trabecular and cortical bone of the femur. Moreover, due to the compensatory increase in intestinal and renal tubular Ca uptake in animals on a low-Ca-diet, the difference in the absolute and relative femur Ca content after four weeks of dietary intervention was only 10%. The net femur Ca gain was about one third lower with a low-Ca-diet. We neither observed correlations of δ^44/42^Ca_Serum_ with the absolute bone Ca content or with the bone Ca gain, or with bone mineral density findings by µCT.

At present, it is unknown to what extent our findings are applicable to humans, and this is a significant future investigation to be undertaken. Dietary Ca intake in humans varies considerably; western diets also contain low amounts of Ca and Ca deficiency is prevalent and often persists over years [40,41]. Our findings deserve validation and consideration in Ca balance modeling approaches and the interpretation of δ^44/42^Ca in serum and urine samples.

## 4. Materials and Methods

### 4.1. Animals

Sprague Dawley rats (Janvier Labs, Le Genest-Saint-Isle, France) were housed at the Interfaculty Biomedical Facility (IBF) at Heidelberg University with a daily cycle of 12 h light and 12 h darkness in groups of three or four according to the recommendations of GV-SOLAS. Tap water was supplied ad libitum. The room temperature was maintained at 22 ± 2 °C with a relative humidity of 50–60%. The animals were allowed seven days of adaptation before the onset of experiments at an age of five to six weeks. Animals were assigned to the three experimental groups using weight-ordered randomization. For anesthesia, 100 mg/kg body weight ketamine and 5 mg/kg body weight xylazine were injected intraperitoneally. The study was approved by the respective authorities (Regierungspräsidium Karlsruhe, Germany, 35-9185 81/G-61/17).

### 4.2. Experimental Protocol

A study scheme is given in Figure 5. The mean body weight at the start of the experiment was 184 ± 8 g, 190 ± 11 g and 190 ± 6 g in groups one to three (ANOVA; *p* = 0.38). Group 1 consisted of 12 male and 6 female SD rats who received a standard-diet ad libitum for two weeks to obtain baseline values for comparison with results in group 2 and 3. In group 1, six female rats were studied and compared to male rats (Appendix A) but not included in the further analysis. Group 2 consisted of 20 male Sprague Dawley rats who received a standard-diet ad libitum for six weeks. Two animals dropped out of the trial prematurely, one because of failure to thrive with poor growth (below mean—2SD) and another because of a painful gait. One animal was excluded from the analysis due to δ^44/42^Ca_Serum_ values beyond 2SD. Group 3 consisted of 20 male Sprague Dawley rats, who switched after two weeks on an ad libitum standard-diet to a low-Ca-diet for four weeks ad libitum. The standard-diet (Altromin, Lage, Germany) consisted of 5000 mg/kg Ca (=0.5%), 3000 mg/kg P (=0.3%), 15 µg/kg Vitamin D3, 50.5 g/kg fat, 178.9 g/kg protein. The low-Ca-diet only differed in the Ca content (2500 mg/kg Ca, =0.25%).

Body weight was measured weekly. Body length was measured at sacrifice during anesthesia. Blood was sampled weekly from the tail vein, and also spot urine and spot feces samples were collected. After two, four and six weeks (prior to sacrifice), rats were placed in a metabolic cage (TECNIPLAST S.p.A, Buguggiate, Italy) for 24 h urine and feces collection. Spot and 24 h urine and feces collections alternated weekly. Since food intake varied from day to day within and between animals, but without any trend over time in any group, mean food intake per animal was calculated by the total amount consumed over the diet-specific intervention periods: group 1 over the two weeks of standard-diet, group 2 over the six weeks of standard-diet, and group three over the two weeks of standard-diet followed by four weeks of the low-Ca-diet. At sacrifice, blood sampling was performed in anesthetized rats by heart puncture and urine collection was performed by bladder puncture. The tibia and femora of both hind legs and the right caudal rib were collected and the adjacent soft tissue was meticulously removed. Bones samples were stored in 70% ethanol at 4 °C for histological analyzes, native bone at room temperature and −25 °C, respectively for Ca studies.

### 4.3. Methods

Urine, food and fecal Ca content were measured by mass spectrometry. Measured food Ca content was as given by the manufacturer. The water intake was estimated based on previously studied age related fluid intake [42], since valid measurements could not consistently be obtained due to losses from the drinking device in the cage. The Ca content of the drinking water was 110 mg/L as given by the Stadtwerke Heidelberg [43]. Serum creatinine, alkaline phosphatase and phosphate were measured according to standard clinical methods (central laboratory, university Hospital Heidelberg).

#### 4.3.1. Bone Ca Quantification

Total Ca analysis was performed using flame atomic absorption spectrometry (F-AAS). Femur samples were weighed, destroyed in 500 µL HNO_3_ at 60 °C for 24–48 h and water was added to increase the final volume to 5 mL. The clear digests were then diluted 1/5000 in a 0.1% La(NO_3_)_3_ solution and applied to the F-AAS instrument. Results are expressed as Ca per wet weighted tissue (mg/g). One animal each from group 2 and one from group 3 was excluded from the Ca content analyses due to technical issues.

#### 4.3.2. Computed Tomography Imaging

Computed tomography (CT) imaging was performed using a micro-CT scanner (Siemens Medical Solutions, Knoxville, TN). The scanning parameters were an X-ray voltage of 60 kV with an anode current of 250 µA and an exposure time of 1000 ms. 360 rotation steps were performed resulting in a total rotation of 360°. The field of view was set to 31.12 mm × 42.10 mm and the binning factor was 1. Under med-high magnification the effective pixel size was 14.30 μm and the protocol was Hounsfield Unit (HU) calibrated. HU were standardized relative to distilled water and air with cylinder size adapted to the specimen size. For image reconstruction, a downsample factor of 2 was used. Analysis of the reconstructed images was performed using the vendor software package Inveon™ Research Workplace (IRW) software 2.2.

Measurements were performed in the right tibiae using predefined regions of interest (ROI), i.e., in the middle of the tibia (=diaphysis), 3/4 of the distance between the measurement point of the diaphysis and the proximal end of the bone (=metaphysis) and 3/4 of the distance between the measuring point of metaphysis and the proximal end of the bone (=epiphysis) as well as for the whole tibia. The thresholds for determining the volumes and density values were defined as follows: “whole bone” 675–5900 HU, “cortical bone” 1800–5900 HU, “cancellous bone” 675–1800 HU.

#### 4.3.3. Bone Histology

The method for quantitative histomorphometry of bone has been described before [44,45]. After careful removal, bones were fixed in ethanol 70% and subsequently embedded in a methylmethacrylate resin. Undecalcified 5 μm thick sections were cut used a Microm HM360 and followed by Goldner-stained for quantitative histomorphometry to determine static bone parameters. At least 1.5 mm² of tissue area was measured per sample. All results are reported as measurements in two dimensions using nomenclature established by the American Society for Bone and Mineral Research [46]. Histological bone analysis was performed in the Laboratory of Pathophysiology of the University of Antwerp, Belgium, using a semi-automatic image analysis program (AxioVision v 4.51, Zeiss, Germany) running a custom program. The different compartments of bone were marked by the operator based on the Goldner stain and the system calculated areas and perimeters. No automatic thresholding was involved. Key parameters that were assessed included bone area per tissue area (B.Ar/T.Ar), osteoid area per bone area (O.Ar/B.Ar) (%), osteoid width (µm), osteoid perimeter per bone perimeter (0.Pm/B.Pm), perimeter of active osteoblasts per osteoid perimeter (Ob.Pm/O.Pm) (%), perimeter of active osteoclasts on eroded perimeter (Oc.Pm/E.Pm) (%), eroded perimeter on bone perimeter (E.Pm/B.Pm). Fibrosis was scored as present or absent. Osteoid seams less than 2 μm in width were not included in primary measurements of osteoid width or area. All parameters studied are figured. All samples for the current study were analyzed in batch, by the same operator, using the same methodology and equipment.

#### 4.3.4. Ca Isotope Determination

##### Sample Digestion and Separation

All samples were processed and compared with one procedural blank and one of each internationally available reference materials (NIST SRM 915a, NIST SRM 1486, IAPSO Standard Seawater) as well as in-house standards (for urine: AK1, for serum: SERA-1). Before isotope ratio measurements were taken, the samples had to be pre-treated. Blanks, reference materials, in-house standards and samples were always treated the same way.

Digestion: First, the organic material needed to be destroyed as thoroughly as possible to release the mineral components (such as Ca). In the case of liquid materials, 250 µL for serum samples and SERA-1, 1 mL for urine samples and AK-1, 60 µL for IAPSO seawater standard, 60 µL for NIST SRM 1486 of a stock solution prepared by dissolving the powder of SRM 1486 were used. In the case of solid samples like feces and food, we weighed an amount of the 10–50 mg sample and dissolved it in a 15 mL PFA beaker with 5mL of nitric acid (HNO_3_, ~14 mol/L) overnight on the hot plate at 120 °C. Bones were pre-treated for approx. 4 h in a 4% sodium hypochlorite solution (NaClO) to remove organics and were afterwards rinsed several times with ultrapure water and dried at 105 °C. Precleaned bone samples were then either drilled at predefined spots (diaphysis, metaphysis and epiphysis, see Section 2.5) and the drilled material was weighed, or the total bone was weighed. In both cases the weighed material was dissolved as described above for food or feces.

After the dissolution of samples, the beakers were opened and the samples were evaporated to dryness at 120 °C. The dried material was then redissolved in 2 mL of 14 mol/L nitric acid.

Dissolved/liquid samples were mixed with a mixture of 8 mL of concentrated nitric acid (HNO_3_) and 2 mL H_2_O_2_ (30%) in PFA tubes, and were then digested in a laboratory microwave (CEM MARS6) following a defined heating program. Next, sample solutions were transferred to 15 mL PFA beakers and dried down at 120 °C and redissolved in 1 mL of 2 mol/L HNO_3,_ and the Ca concentration of these solutions was determined using Q-ICP-MS (Agilent 7500cx, Agilent Technologies, Santa Clara, CA, USA).

Separation: The second step of the chemical treatment is the chromatographic separation of Ca from other elements like Na, K, Mg, Sr, etc. An aliquot of sample containing 5 µg to 25 µg Ca is purified using an automatic purification system (prepFast MC, ESI, USA) following a defined procedure [47]. The collected Ca-fraction is dried down and then redissolved in a mixture of 1 mL of concentrated HNO_3_ and 0.5 mL H_2_O_2_ 30% and placed on a hotplate with closed lids at 120 °C for 4 h. Afterwards, the beakers were opened and the solution was dried down at 120 °C and then re-dissolved in an appropriate volume of 0.2 mol/L HNO_3_ to yield a concentration of 2.5 mg/L Ca. These solutions are further used for the measurements of the isotopic ratios.

##### Mass Spectrometry and Data Evaluation

Ca isotope measurements were performed on a MC-ICP-MS (Neptune plus, Thermo Fisher Scientific, Bremen, Germany) at the mass spectrometer facilities of the GEOMAR Helmholtz Center for Ocean Research Kiel, Germany. The mass spectrometer was equipped with nine Faraday cups, of which eight are moveable. The mass spectrometer was set up to measure masses 42, 43, 43.5 and 44 simultaneously. In order to suppress interfering Ca- and Ar-hydrides (e.g., ^40^Ar^1^H_2_ on ^42^Ca) an APEX IR (ESI, Omaha, NE, USA) sample introduction system was used. All measurements were performed in medium resolution (MR, m/Δm ~4000) on the interference-free plateau of the low mass side of the peaks. This was achieved by choosing an appropriate center cup mass which was verified on a daily basis [48]. Instrumental fractionation (mass bias) was corrected by applying the standard-sample-bracketing approach. The measurement of a sample was bracketed by measurements of a ~2.5 μg/mL Ca solution prepared from a 10,000 μg/g Ca ICP reference solution (Ca ICP). Every sample solution was measured at least four times per session and the mean value was used for further calculations. The Ca isotopic composition is reported as δ^44/42^Ca in parts per mill (‰).

Further tests confirmed that there is negligible isotope fractionation during purification (<0.01‰). Although nearly all Sr was removed from the samples during chemical preparation, we corrected samples for doubly charged Sr (^84^Sr^++^, ^86^Sr^++^ and ^88^Sr^++^) that can interfere with the measurement of ^42^Ca^+^, ^43^Ca^+^ and ^44^Ca^+^, as they have the same mass to charge ratio by measuring ^87^Sr^++^ [15]. Following Morgan et al. [15], a single measurement was rejected when |δ^44/42^Ca—2·δ^43/42^Ca| > 0.2‰. A sample measurement (average of 4 single measurements) was rejected when the average intensity was outside a 70–130% intensity window compared to the average intensity of the NIST SRM 915a measurements from the same batch. A complete session was rejected when more than one of the measured reference materials deviated >0.2‰ from the literature value or the data did not fall along the mass-dependent fractionation line. δ^44/42^Ca of international standard materials were in line with published values.

The Ca isotopic composition was reported as δ^44/42^Ca relative to NIST SRM 915a in parts per thousand (‰) following Coplen (2011):δ^44/42^Ca = [(^44^Ca/^42^Ca)_Sample_ − (^44^Ca/^42^Ca)_NIST SRM 915a_]/(^44^Ca/^42^Ca)_NIST SRM 915a_(1)

Based on the notation above, Ca isotope values turn negative, because (^44^Ca/^42^Ca)_Serum_ values are lower than the (^44^Ca/^42^Ca)_NIST SRM 915a_ values. Values are positive when (^44^Ca/^42^Ca)_Sample_ values are greater than the (^44^Ca/^42^Ca)_NIST SRM 915a_ value. Note that the sign of the δ^44/42^Ca-notation also points into the direction of fractionation. Negative δ^44/42^Ca values indicate enriched ^42^Ca, positive values indicate ^44^Ca enrichment.

### 4.4. Statistics

For statistical analysis Graph Pad Prism 9.0 (La Jolla, CA, USA) and Microsoft Excel (Redmond, WA, USA) were used. Normal distribution was assessed graphically and by Shapiro-Wilk test. Normally distributed data were reported as means and standard deviation (SD). Outliers were defined as values beyond 2SD. Means were tested with an unpaired two-sided Student’s *t*-test and one- and two-factor ANOVA with repeated measures. Non-normal variables were summarized with medians, (interquartile range) and comparison of median values between two/three groups was conducted using a two-sided Mann-Whitney, respectively. For quantification of linear correlations, a Pearson correlation coefficient was applied. The significance level p for α was set at <0.05.

## 5. Conclusions

In summary, our longitudinal analysis of total body Ca balance and δ^44/42^Ca in the diet, feces urine and in bone revealed several important findings. δ^44/42^Ca_Bone_ is lower in the diaphysis, the site of the highest Ca deposition, than in the epiphysis and metaphysis, independent of the dietary Ca supply. Bone distribution of δ^44/42^Ca is independent of the mechanical stress the bone is exposed to, and is similar in female and male rats. Dietary Ca restriction in rapidly growing rats results in subtotal intestinal Ca absorption, a marked reduction in δ^44/42^Ca in feces and urine, and in an increase in δ^44/42^Ca_Serum_, suggesting dietary Ca dependent differences in intestinal and renal tubular Ca isotope balances. δ^44/42^Ca_Serum_ inversely correlates with the intestinal Ca uptake and at time of sacrifice inversely with histological features of bone mineralization, but not with dietary Ca dependent bone Ca content and bone Ca gain. Our findings deserve validation in humans and inform future mathematical multi-compartmental modelling of δ^44/42^Ca.

## Figures and Tables

**Figure 1 ijms-23-07796-f001:**
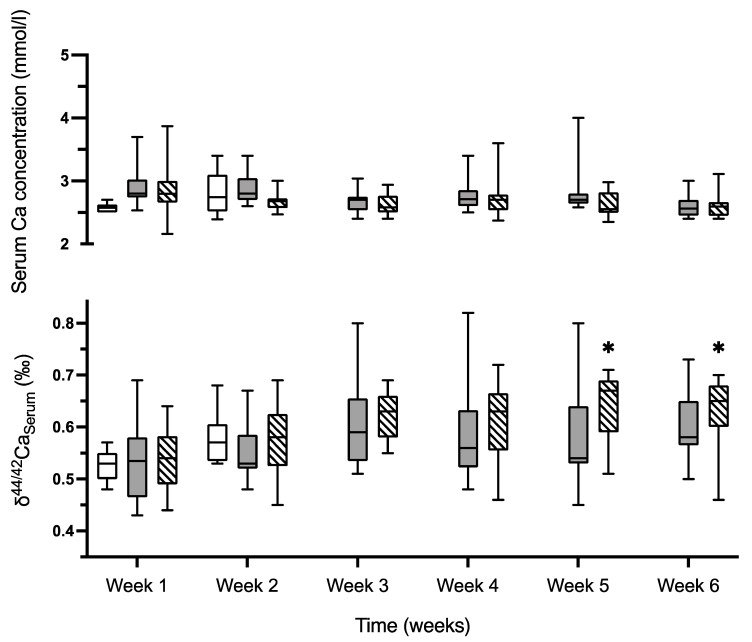
Serum Ca concentrations and δ^44/42^Ca_Serum._ Serum Ca concentrations (upper *y*-axis) and δ^44/42^Ca_Serum_ (lower *y*-axis) were measured weekly in the baseline group (0.5% Ca diet for two weeks, empty boxes, n = 9), the standard-diet group (six weeks 0.5% Ca diet, gray boxes, n = 13) and the low-Ca-diet group (two weeks 0.5% Ca diet followed by four weeks of 0.25% Ca diet, hatched boxes, n = 13). Minimal and maximal values (whiskers), the interquartile range (box) and the median (solid line) are plotted. Serum Ca concentrations were similar in all three groups throughout the study, δ^44/42^Ca_Serum_ increased in animals on low-Ca-diet (ANOVA, *p* = 0.007), reaching significantly higher values at week five and six (ANOVA, * *p* < 0.05 versus week two).

**Figure 2 ijms-23-07796-f002:**
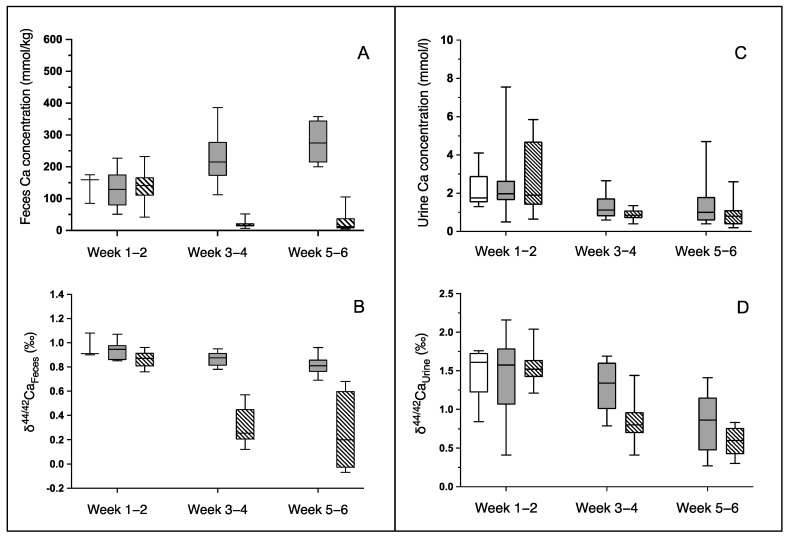
Fecal and urine Ca concentrations, δ^44/42^Ca_Feces_ and δ^44/42^Ca_Urine_ in rats after two weeks on standard-diet (empty boxes) and subsequent four weeks on standard- (gray boxes) and low-Ca-diet (hatched boxes). Fecal Ca concentrations (**A**) and δ^44/42^Ca_Feces_ (**B**) were lower in animals on low-Ca-diet than in animals on standard-diet (*p* = 0.005 and *p* < 0.0001). Urine Ca concentrations (**C**) and δ^44/42^Ca_Urine_ (**D**) were lower in animals on low-Ca-diet (ANOVA, *p* = 0.03 and *p* < 0.0001). Relatively more heavy Ca isotopes were absorbed in the intestine and the renal tubular system in low-Ca-diet rats.

**Figure 3 ijms-23-07796-f003:**
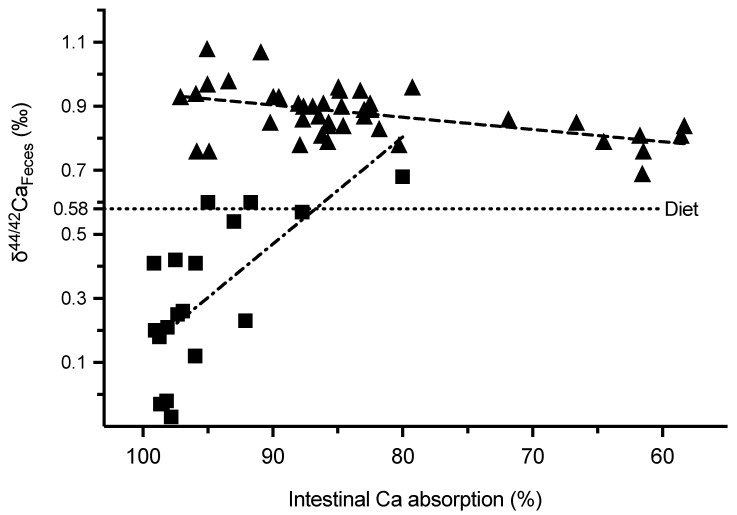
Relative intestinal Ca absorption versus δ^44/42^Ca_Feces._ The amount of Ca absorbed relative to the dietary Ca intake is given on the *x*-axis and δ44/42CaFeces on the *y*-axis in animals on standard-Ca-diet (triangles; baseline values, group 2 weeks one to six and group 3 weeks one to two) and low-Ca-diet (squares; group 3 week three to six). The dotted line marks the isotope composition of the diet. In all animals on the standard-diet, increasing intestinal Ca absorption was accompanied by an increase in δ^44/42^Ca_Feces_ (pearsons correlation, r = 0.50, *p* = 0.0006; n = 43) up to about 95% absorption. In contrast, in animals on a low-Ca-diet relative intestinal Ca absorption showed values below their corresponding δ^44/42^Ca_Diet_-value and correlated inversely (pearsons correlation, r = −0.68, *p* = 0.0015; n = 19).

**Figure 4 ijms-23-07796-f004:**
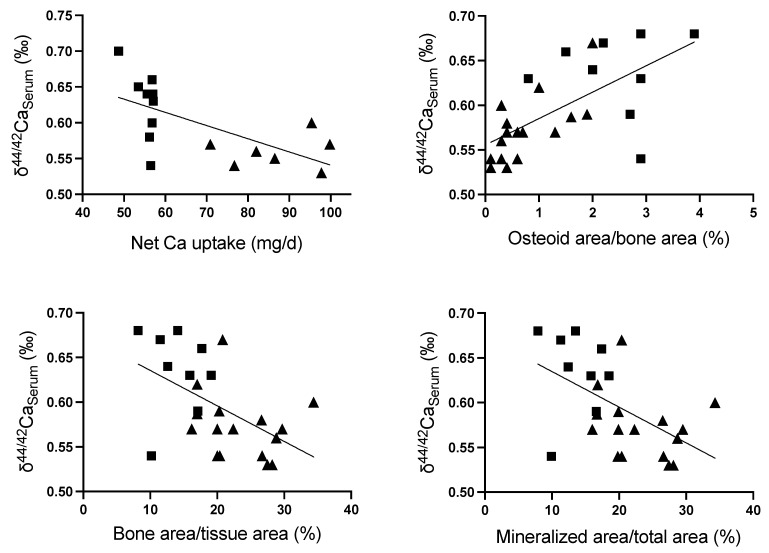
δ^44/42^Ca_Serum,_ net intestinal Ca uptake and bone histology findings. δ^44/42^Ca_Serum_ in rats on standard-diet (triangles; baseline values, group 2 weeks three to six) and low-Ca-diet (squares; group 3, weeks three to six) are shown relative to net intestinal Ca uptake and key bone histology parameters. Mean weekly δ^44/42^Ca_Serum_ during weeks three to six inversely correlated with the net intestinal Ca uptake during this period (pearsons correlation, r = −0.62, *p* = 0.007; n = 16). At sacrifice, δ^44/42^Ca_Serum_ correlated with osteoid area/bone area (pearsons correlation, r = 0.64, *p* = 0.0005, n = 25) and negatively with bone area/tissue area (pearsons correlation, r = −0.53; *p* = 0.007, n = 25) and mineralized area/total area (pearsons correlation, r = −0.53, *p* = 0.006; n = 25).

**Figure 5 ijms-23-07796-f005:**
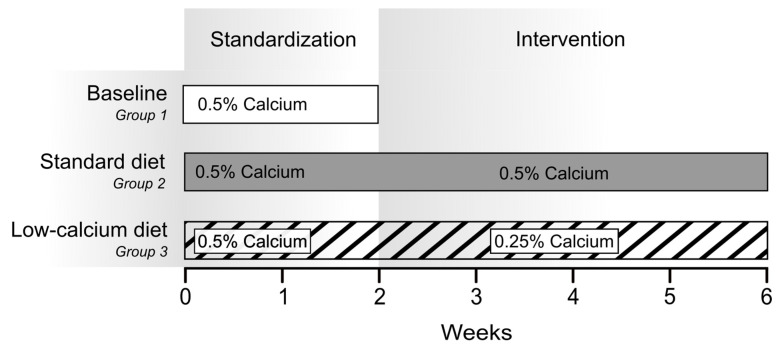
Study experimental organization. All animals received the standard-diet for two weeks. Animals of group 1 (n = 18, empty bar) were sacrificed to obtain baseline values. Animals in group 2 were maintained on standard-diet (n = 20, gray bar), animals in group 3 were switched to a low-Ca-diet (n = 20, hatched bar). Blood samples were taken weekly. Spot urine and feces were collected after one, three and five weeks. 24 h urine and feces were collected after two, four and six weeks.

**Table 1 ijms-23-07796-t001:** Ca balances measured during two weeks of standard-diet (group 1, 0.5% Ca) followed by four weeks on standard (group 2, 0.5% Ca) or low-Ca-diet (group 3, 0.25% Ca).

		Baseline (Group 1, n = 12)	Standard-Ca-Diet (Group 2, n = 19)	Low-Ca-Diet (Group 3, n = 20)	*p*-Value ^a^
Week 1–2	Food intake (g/day) Dietary Ca intake (mg/day)	22.1 ± 1.0	22.3 ± 1.7	21.5 ± 0.4	0.44
		110.6 ± 5.0	111.5 ± 8.3	107.4 ± 2.1	0.44
	Fluid intake (mL/day)	28.8 ± 1.0	29.4 ± 1.5	29.8 ± 1.2	0.38
	Fluid Ca intake (mg/day)	3.2 ± 0.1	3.2 ± 0.2	3.3 ± 0.1	0.38
	Feces (g/day) Fecal Ca loss (mg/day)	2.8 ± 0.7	2.7 ± 1.0	3.0 ± 1.0	0.76
		12.9 ± 6.4	12.9 ± 9.7	15.2 ± 5.0	0.77
	Intestinal Ca absorption ^b^ (%)	88.6 ± 5.6	88.8 ± 8.5	86.2 ± 4.5	0.68
	Intestinal Ca absorption (mg/day)	100.9 ± 6.4	101.8 ± 9.7	95.4 ± 5.4	0.18
	Urine excretion (mL/day) Urine Ca excretion (mg/day)	8.5 ± 3.2	7.9 ± 2.8	10.2 ± 3.3	0.08
		0.7 ± 0.4	1.1 ± 0.9	0.8 ± 0.7	0.47
	Net body Ca uptake ^c^ (mg/day)	100.4 ± 6.6	100.8 ± 9.8	94.6 ± 4.9	0.99
Week 3–4	Food intake (g/day) Dietary Ca intake (mg/day)		22.3 ± 1.7	21.5 ± 0.7	0.40
			111.5 ± 8.3	53.7 ± 1.9	<0.0001
	Fluid intake (mL/day)		38.7 ±1.6	39.6 ± 2.2	0.10
	Fluid Ca intake (mg/day)		4.3 ± 0.2	4.4 ± 0.2	0.10
	Feces (g/day) Fecal Ca loss (mg/day)		2.8 ± 1.1	3.2 ± 1.2	0.23
			25.9 ± 16.2	2.7 ± 2.0	0.0003
	Intestinal Ca absorption ^b^ (%)		76.6 ± 14.4	95.3 ± 3.4	0.0009
	Intestinal Ca absorption (mg/day)		88.7 ± 16.7	55.3 ± 2.0	<0.0001
	Urine excretion (mL/day) Urine Ca excretion (mg/day)		11.6 ± 6.0	13.9 ± 5.4	0.22
			0.6 ± 0.2	0.4 ± 0.1	0.09
	Net body Ca uptake ^c^ (mg/day)		89.4 ± 16.3	54.9 ± 1.9	<0.0001
Week 5–6	Food intake (g/day) Dietary Ca intake (mg)		22.3 ± 1.7	21.5 ± 0.7	0.40
			111.5 ± 8.3	53.7 ± 1.9	<0.0001
	Water intake (mL/day)		46.1 ± 1.6	446.3 ± 2.9	0.67
	Fluid Ca intake (mg/day)		5.1 ± 0.2	5.1 ± 0.3	0.67
	Feces (g/day) Fecal Ca loss (mg/day)		2.1 ± 0.9	2.6 ± 1.5	0.30
			25.3 ± 13.5	2.7 ± 3.7	0.0002
	Intestinal Ca absorption ^b^ (%)		78.3 ± 11.6	95.3 ± 6.2	0.001
	Intestinal Ca absorption (mg/day)		91.2 ± 13.5	56.1 ± 3.7	<0.0001
	Urine excretion (mL/day) Urine Ca excretion (mg/day)		17.2 ± 5.8	13.2 ± 6.9	0.14
			1.0 ± 0.9	0.3 ± 0.3	0.001
	Net body Ca uptake ^c^ (mg/day)		90.6 ± 13.5	55.7 ± 3.6	<0.0001
Week 1–6	Net body Ca uptake (mg/6 weeks)		3931	2872	
Week 3–6	Net body Ca uptake (mg/4 weeks)		2520	1548	<0.0001

^a^ = week 1/2: One-way ANOVA; week three/four and week five/six: *t*-test; ^b^ = food- and water-Ca intake minus fecal Ca excretion; ^c^= food- and water-Ca intake minus fecal and urine Ca excretion.

**Table 2 ijms-23-07796-t002:** Femur Ca content after two weeks on standard-diet (group 1, 0.5% Ca) and a subsequent four weeks of standard-diet (group 2) and low-Ca-diet (group 3, 0.25% Ca).

	Baseline Week 2 (Group 1; n = 6)	Standard-Ca-Diet Week 6 (Group 2, n = 19)	Low-Ca-Diet Week 6 (Group 3, n = 19)	*p*-Value ^a^	
				G1 vs. G2	G2 vs. G3
Femur length (mm)	32.3 ± 0.4	37.5 ± 0.3	38.1 ± 0.9	<0.0001	0.17
Femur Ca content (mg/femur)	81.8 ± 4.5	121 ± 15	109 ± 16	<0.0001	0.02
Relative Ca content (mg/g bone)	97.4 ± 5.3	104 ± 12	92.9 ± 14	0.21	0.03
∆ Ca gain femur (mg)		39.6 ± 15	27.0 ± 16		0.02

^a^ = *t*-test.

**Table 3 ijms-23-07796-t003:** Tibia histomorphometry after two weeks on standard-diet (group 1, 0.5% Ca) and a subsequent four weeks of standard-diet (group 2, 0.5% Ca) and low-Ca-diet (group 3, 0.25% Ca). Bone analysis was performed in the proximal metaphysis and epiphysis.

Left Tibia	Baseline Week 2 (Group 1, n = 12)	Standard-Ca-Diet Week 6 (Group 2, n = 11)	Low-Ca-Diet Week 6 (Group 3, n = 12)	*p*-Value ^a^
Bone area/Tissue area (%)	22.5 ± 6.4	20.4 ± 5.2	14.0 ± 3.2	0.002
Trabecular number (/mm)	5.6 ± 1.2	4.1 ± 0.6	3.2 ± 0.7	0.003
Trabecular separation (mm)	136 (96–184)	170 (150–209)	255 (209–288)	0.003 ^b^
Trabecular thickness (mm)	51 ± 4.4	62 ± 8.6	57 ± 5.8	0.09
Mineralized area/ total area (%)	22.4 ± 6.4	20.2 ± 5.2	13.7 ± 3.1	0.001
Osteoid area/ bone area (%)	0.5 (0.2–1.1)	0.6 (0.4–0.9)	2.2 (1.6–2.9)	<0.0001 ^b^
Osteoid perimeter/ Total perimeter (%)	1.4 (0.7–2.5)	4.1 (2.7–6.1)	13 (9.4–15)	<0.0001 ^b^
Osteoid width (µm)	4.9 ± 1.5	4.5 ± 0.7	4.9 ± 1.3	0.41
Eroded parameter/ Total parameter (%)	6.9 ± 3.7	5.3 ± 3.0	4.8 ± 3.2	0.64
Osteoblast perimeter/ Osteoid perimeter (%)	25.1 ± 28.5	19.3 ± 20.7	37.7 ± 23.0	0.06
Osteoblast perimeter/Total perimeter (%)	0.4 (0.0–1.5)	0.7 (0.00–1.4)	4.2 (1.8–8.3)	0.0002 ^b^
Osteoclast perimeter/ Eroded perimeter (%)	24.1 ± 12.1	17.6 ± 10.4	22.1 ± 13.7	0.39
Osteoclast perimeter/ Total perimeter (%)	1.7 ± 1.1	1.1 ± 0.0	1.2 ± 0.8	0.82

^a^ = group 2 vs. group 3, *t*-test; ^b^ = Mann-Whitney-test.

**Table 4 ijms-23-07796-t004:** ^δ44/42^Ca_Bone_ of the femora and the lowest ribs (1) and of the left tibia epiphysis, metaphysis and diaphysis (2) of rats after two weeks on standard-diet (group 1, 0.5% Ca) followed by four weeks of standard-Ca-diet (group 2, 0.5% Ca) or low-Ca-diet (group 3, 0.25% Ca).

(1)							
δ^44/42^Ca_Bone_ (‰)			Baseline Week 2 (Group 1, n = 9)	Standard-Ca-Diet Week 6 (Group 2, n = 6)	Low-Ca-Diet Week 6 (Group 3, n = 10)	*p*-Value	
						G1 vs. G2 ^a^	G2 vs. G3 ^a^
		Weight-bearing bone (femora)	0.15 ± 0.05	0.24 ± 0.02	0.29 ± 0.03	0.0008	0.01
		non-weight-bearing bone (lowest rib)	0.13 ± 0.05	0.24 ± 0.01	0.29 ± 0.03	0.0003	0.002
*p*-value ^a^			0.61	0.89	0.75		
**(2)**							
		Diaphysis (tibia)	0.18 ± 0.06	0.21 ± 0.09	0.28 ± 0.03	0.71	0.05
		Metaphysis (tibia)	0.27 ± 0.07	0.26 ± 0.07	0.33 ± 0.03	0.67	0.006
		Epiphysis (tibia)	0.26 ± 0.04	0.24 ± 0.07	0.32 ± 0.03	0.71	0.009
*p*-value ^a^		Diaphysis vs. Metaphysis	0.016	0.37	0.0006		
		Diaphysis vs. Epiphysis	0.007	0.54	0.008		
		Metaphysis vs. Epiphysis	0.55	0.74	0.40		

^a^ = *t*-test.

## Data Availability

All data in this study is presented within the article and Appendix A.

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
