# Peer review of "Nutritional Calcium Supply Dependent Calcium Balance, Bone Calcification and Calcium Isotope Ratios in Rats"

_ijms, 2022, doi:10.3390/ijms23147796_

Round 1

Reviewer 1 Report

The manuscript is badly formated, not follow the journal structure

line 37, replace we with "the study...."

line 64, don't start with 2.1%, reformulate

add citation in line 72 after "isotopes"

Lines 83, 252 adjust the unit and delete the extra brackets, 

what do you mean with c.f.?

check the language mistakes as in line 106 "we, therefore... check all manuscript

line 108 it should be materials

line 169, add the abbreviation of CT

line 167, adjust the unit into mg/g Ca

lines 226,227 adjust the font

line 231, adjust citation

center the equation in line 267

reformulate the head of all tables 

supplementary materials should be after references

references are outdated please update

check the output of all references and follow the journal instruction

line 603, separate the conclusion in section 

Reviewer 2 Report

The authors present their results of experiments mnapping the links between the bone histology, bone calcification, and Ca isotopes in rat. The study brings some novelty and in general, it would deserve to be published. However, several improvements remain to be done to make the paper more beneficial to the Journal readers, namely:

-In Abstract, some results are presented including p-values, some are not. Be consistent. What do yo umean by "slightly increased" (line 29)? THe conclusion of the abstract shold be written in simple past. The sentence on line 37 might be rephrased.

-What dou you mean by "educt" on line 70?

-Formulate clearly testable biological hypothesis/hypotheses at the end of the Introduction. "We studied" (line 106) is not specific enough.

- I was missing any the details on the type of microtome in section 2.3.3. Moreover, how many sections per tissue blocks and per animal were evaluated? How were the sections selected and sampled? What were the thresholding criteriawhen using the semi-automatic image analysis? What were the preprocessing steps? All these details can make your study more reproducible.

-Were there any internal controls in the study?

-In my opinion, complete morphometric data should be published along with the statistics, graphs, and tables.

-I am quite concerned about the morphometrics. Why did you use two-dimensional morphometric tools when three-dimensional data were available frmo the micro-CT? Also, I am missing any discussion regarding the bone design-based stereology as this is the gold standard when it comes to morphometry. Try to compare your methods with those of design-based stereology. perhaps you can discuss the meaning of the parameters? Please provide a list of morphometric parameters you used and link the parameters with their biological meaning and interpretation. This would make your data much more readable, comparable, and citable.

-The equation on the line 267 seems not to be aligned.

-2.4. Statistics - did you detect some data with non-parametric distribution?

-In FIgure 1, the meaning of the box and whiskers should be explained in the figure legend. Are these medians, quartiles, min-max range?

-The Discussion is really lengthy. Consider making it more succint. Split it into more subsections, each addressing the biological hypothesis formulated in the Aims.

Round 2

Reviewer 1 Report

Now can be accepted in IJMS